



# Soils from cold and snowy temperate deciduous forests release more nitrogen and phosphorus after soil freeze-thaw cycles than soils from warmer, snow-poor conditions

Juergen Kreyling[1], Rhena Schumann[2], Robert Weigel[1,3]

[1]Experimental Plant Ecology, University of Greifswald, Greifswald, D-17489, Germany
[2]Biological Station Zingst, Applied Ecology & Phycology, University of Rostock, 18374 Zingst, Mühlenstraße 27
[3]Albrecht-von-Haller-Institute for Plant Sciences, University of Goettingen, Goettingen, 37073, Germany

*Correspondence to*: Juergen Kreyling (juergen.kreyling@uni-greifswald.de)

**Abstract.** Effects of global warming are most pronounced in winter. A reduction in snow cover due to warmer atmospheric temperature in formerly cold ecosystems, however, could counteract an increase in soil temperature by reduction of insulation. Thus, soil freeze-thaw cycles (FTC) might increase in frequency and magnitude with warming, potentially leading to a

disturbance of the soil biota and release of nutrients.

Here, we assessed how soil freeze-thaw magnitude and frequency affect short-term release of nutrients in temperate deciduous forest soils by conducting a three factorial gradient experiment with ex-situ soil samples in climate chambers. The fully-crossed experiment included soils from forests dominated by *Fagus sylvatica* (European beech) that originate from different winter climate (mean coldest month temperature range $\Delta T > 4$ K), a range of FTC magnitudes from no (T = 4.0 °C) to strong (T = -

11.3 °C) soil frost, and a range of FTC frequencies (f = 0–7). We hypothesized that higher frost magnitude and frequency, respectively, will increase the release of nutrients. Furthermore, soils from cold climates with historically stable winter soil temperatures due to deep snow cover will be more responsive to FTC than soils from warmer, more fluctuating winter soil climates.

FTC magnitude and, to a lesser extent, also FTC frequency resulted in increased nitrate, ammonium, and phosphate release

almost exclusively in soils from cold, snow-rich sites. The hierarchical regression analyses of our three-factorial gradient experiment revealed that the effects of climatic origin (mean minimum winter temperature) followed a sigmoidal curve for all studied nutrients and was modulated either by FTC magnitude (phosphate) or by FTC magnitude and frequency (nitrate, ammonium) in complex two- and, for all studied nutrients, in threefold interactions of the environmental drivers. Compared to initial concentrations, soluble nutrients were predicted to increase to 250 % for nitrate (up to 16 µg $NO_3$-N kg-1DM), to

110 % for ammonium (up to 60 µg $NH_4$-N kg-1DM), and to 400 % for phosphate (2.2 µg $PO_4$-P kg-1DM) at the coldest site for strongest magnitude and highest frequency. Soils from warmer sites showed little nutrient release and were largely unaffected by the FTC treatments except for above-average nitrate release at the warmest sites in response to extremely cold FTC magnitude.





We suggest that currently warmer forest soils have historically already passed the point of high responsiveness to winter
climate change, displaying some form of adaptation either in the soil biotic composition or in labile nutrient sources. Our data
suggests that previously cold sites, which will lose their protective snow cover during climate change, are most vulnerable to
increasing FTC frequency and magnitude, resulting in strong shifts in nitrogen and phosphorus release. In nutrient poor
European beech forests of the studied Pleistocene lowlands, nutrients released over winter may be leached out, inducing
reduced plant growth rates in the following growing season.

## 1 Introduction

Climate is warming over-proportionally in northern latitudes and during winter (IPCC, 2013). This has potentially important
consequences for nutrient cycling and ecosystem functioning (Kreyling, 2020). Cold-temperate deciduous forests are
experiencing more fluctuating soil temperatures and potentially also more frequent soil freeze-thaw cycles (FTC) because
reduced or completely missing snow cover exposes them to strongly fluctuating air temperatures (Kreyling, 2020). These
forests are nutrient limited (Bontemps et al., 2011) and soil nutrient release in response to FTC frequency and FTC magnitude
of forests differing in their past and present climate are therefore of high ecological and economical importance.

### 1.1 *Winter climate change in the temperate deciduous forests of Central Europe*

Winters in temperate regions are projected to become warmer, more variable, and wetter with precipitation increasing and
changing from snow to rain (Stocker, 2014; Yang and Christensen, 2012). Largest decreases in snowfall are expected for
regions with winter mean air temperatures ranging from -5 to +5 °C, while colder regions (boreal, arctic) might even receive
increased snowfall (Brown and Mote, 2009; Scherrer and Appenzeller, 2006). The shift from snow to rain drastically reduces
soil insulation and exposes soils to the fluctuations of air temperatures (Groffman et al., 2001). While insulation by snow can
prevent soil freezing even in boreal climates (Isard and Schaetzl 1998), missing snow can lead to increased soil frost in regions
with sustained air frost (Groffman et al., 2001; Brown and DeGaetano, 2011; Henry, 2008), increased frequency of soil freeze
thaw cycles (FTC) in regions where air temperatures fluctuate around 0 °C (Henry, 2008; Campbell et al., 2010), or reduced
soil frost where even minimum air temperatures rarely drop below the freezing point (Kreyling and Henry, 2011).

### 1.2 *Ecological consequences of altered soil temperatures*

Many relevant ecological processes are driven by winter soil temperatures such as activity and survival of organs and
organisms (Kreyling, 2010; Campbell et al., 2005). Soil freezing represents an important threshold for microbial activity
because of reduced availability of liquid water (Mikan et al., 2002). However, colder temperatures are typically required to
cause microbial lysis as microbial growth can continue below freezing (McMahon et al., 2009). Sub-lethal effects of freezing
on soil microorganisms are not well understood, and the length of freezing, the number of freeze thaw cycles, and the rate of
freezing can all increase cell damage for a given freezing intensity (Elliott and Henry, 2009; Vestgarden and Austnes, 2009).





In addition, soil microorganisms which survive freezing and desiccation can be lethally damaged via osmotic shock upon
exposure to melt water (Jefferies et al., 2010) and physiological re-adaptation to thawing conditions may lead to microbial
carbon and nutrient release (Schimel et al., 2007). Consequently, soil freezing can disrupt soil microbial activity (Bolter et al.,
2005; Yanai et al., 2004) and affect key microbial processes such as ammonification, nitrification and denitrification (Urakawa
et al., 2014; Watanabe et al., 2019; Hosokawa et al., 2017). Furthermore, soil freezing can damage plant roots (Tierney et al.,
2001; Reinmann and Templer, 2018; Kreyling et al., 2012a; Weih and Karlsson, 2002), induce soil nitrogen (N) leaching
(Joseph and Henry, 2009; Matzner and Borken, 2008), increase soil trace gas losses (Reinmann and Templer, 2018; Matzner
and Borken, 2008), reduce N uptake by trees (Campbell et al., 2014), decrease plant productivity (Göbel et al., 2019;
Comerford et al., 2013; Reinmann et al., 2019) and can ultimately lead to plant mortality (Schaberg et al., 2008; Buma et al.,
2017). In addition to direct frost damage, the listed consequences of soil freezing on plant performance are commonly
explained by altered nutrient, mainly N and P, availabilities (Kreyling, 2020). Freezing can also affect release of these nutrients
by physically breaking up soil aggregates (Oztas and Fayetorbay, 2003) or organic litter (Hobbie and Chapin, 1996) and by
reducing soil water flow rates (Iwata et al., 2010).

Changes in FTC frequency can affect microbial communities, e.g. increasing saprotrophic fungal activity (Kreyling et al.,
2012b). Nitrogen leaching from soil columns subjected to FTC remaining high even after 10 FTC further emphasizes the
importance of FTC frequency (Joseph and Henry, 2008). Taken together, FTC can affect soil nutrient release through damage
and lysis of microbial and plant cells, through altered soil biotic activity, and/ or through physical disruption of abiotic and
dead organic particles. In particular for nutrient limited ecosystems, altered occurrence of FTC with climate change could
consequently affect ecosystem functioning.

### 1.3 *Beech forests of Pleistocene lowlands as important and potentially affected ecosystem*

Beech forests are the zonal vegetation of Central Europe and face multiple anthropogenic pressures while still providing vast
ecosystem services (Ammer et al., 2018). Beech (*Fagus sylvatica* L.) naturally dominates all over Central Europe under a wide
range of soil conditions and occurs in regions with less than 550 to more than 2000 mm of annual rainfall on nearly all
geological substrates if drainage is sufficient (Leuschner et al., 2006). Even when growing on marginal soils, beech forests
have a N demand of about 100 kg N ha$^{-1}$ year$^{-1}$ which is several times higher than current atmospheric N-deposition in Central
Europe (Rennenberg and Dannenmann, 2015). Nitrogen availability is consequently still the most limiting factor of beech
growth at marginal as well as at productive sites (Bontemps et al., 2011). N availability is largely determined by internal N
cycling through microbial mineralization and immobilization (Guo et al., 2013). Any alteration in the microbial community
and activity, such as in response to FTC, therefore has the potential to affect nutrient cycling and, thereby, ecosystem
functioning of this ecologically and economically important ecosystem (Simon et al., 2017).

In addition to N limitation, phosphorus (P) nutrition of beech is recently decreasing and could emerge as another limiting factor
for beech growth on nutrient poor soils (Talkner et al., 2015). Implications of climate change on P release of beech forest soils
should therefore also be investigated.



### 1.4 *Hypotheses*

We hypothesized that soil freeze-thaw cycles (FTC) induce nutrient release following saturation curves both with increased FTC magnitude and increased FTC frequency. We expected the combination of FTC magnitude and FTC frequency to be

additive. We further hypothesized that soils from colder macroclimates which are characterized by more persistent and protective snow cover are more responsive in release of nutrients in the face of FTC than soils from warmer sites with more fluctuating winter soil temperatures.

### 2 Materials & methods

The effects of FTC magnitude and FTC frequency on the short-term release of nutrients in temperate deciduous forest soils

was assessed in a three-factorial gradient experiment with *ex-situ* soil samples in climate chambers. The fully-crossed experiment included soils from seven forests dominated by *Fagus sylvatica* (beech) that (1) originate from different winter climate (mean winter minimum temperature range $\Delta T > 4$ K) and were exposed to (2) a range of FTC magnitudes from no (T = 4.0 °C) to strong (T = -11.3 °C) soil frost, and (3) a range of FTC frequencies (f = 0–7).

### 2.1 *Forest sites and soil sample collection*

Soil samples for this study stemmed from seven sites located between Rostock (Germany) and Gdansk (Poland) which are mono-dominated by mature European beech. Along the 500-km study gradient, the sites differ markedly in winter climate with mean average winter air temperatures ($\Delta T = 4.0$ K) and mean minimum winter air temperatures ($\Delta T = 3.8$ K) decreasing towards the east, which over-proportionally drives the differences in mean annual temperature ($\Delta T = 2.8$ K; for details see Table 1). From west to east, mean annual precipitation as snow increases from 50 to 110 mm while annual precipitation is

rather uniform (540 to 630 mm). With respect to winter air temperature differences, the study area is representative of a large part of the temperature range of beech as the major forest tree in Europe, while for summer precipitation, which is considered to be a major driver of beech growth (Hacket-Pain et al., 2018), differences are relatively small (Table 1).

The study sites are located in the Pleistocene lowlands with glacial deposits as bedrock. All sites share the same soil type (sandy Cambisol) and similar soil texture (sandy silt to silty sand). Sites were selected for similar forest stand structure, i.e.

tree height about 30 m (ranging between 27–39 m), tree diameter about 45 cm (ranging between 37–52 cm), and canopy closure 70–80 %. In order to achieve this uniform stand structure, differences in mean tree age across sites was accepted (76-167 years). At each site, we systematically selected the sampling sites in proximity to site-representative target trees. A dendroecological pre-study (Weigel et al., 2018) identified these target tree individuals by selecting for the best correlations between individual tree-ring series and the site chronology (the mean of all individual tree-ring series of a site) during the last

30 years (three target trees out of 20 at all but coldest site, three out of 40 at coldest site). Consequently, the selected target trees within each site showed very similar growth patterns over the past 30 years and ideally represented the growth–





environment relationship of the whole stand. At each site, we randomly selected one target tree and took three soil sub-samples (later on mixed) at a distance of 3 m from the selected individual. Samples were stored at 4 °C for 16 weeks.

The mixed samples per site were carefully homogenized and subsampled to 10 g for the subsequent FTC treatment (see below).

Soil moisture at the start of the FTC treatment ranged between 19.4 and 36.6 % between the sites and was not significantly related to climate at site origin (correlation to mean minimum air temperature: $R^2 = 0.33$, p = 0.103). Initial values for the analyzed nutrients were also recorded at the start of the FTC treatment with the same methodology as described below and are presented in Table 2.

Table 1: Climatic site characteristics for the seven sampled beech forest stands. Tmin_wt: winter mean minimum temperature (°C); MAT: mean annual temperature (°C); Tave_wt: winter (Dec.(prev. yr) - Feb.) mean temperature (°C); Tave_sm: summer (Jun. - Aug.) mean temperature (°C); MAP: mean annual precipitation (mm); PAS: precipitation as snow (mm) between August in previous year and July in current year; Prec_sm: summer precipitation (mm). All climatic data for the reference period 1961–1990 according to "climateEU" 4.63, (Hamann et al., 2013; Wang et al., 2012).

| Site | Longitude | Latitude | Altitude | Tmin_wt | MAT | Tave_wt | Tave_sm | MAP | PAS | Prec_sm |
|---|---|---|---|---|---|---|---|---|---|---|
| BB | 13.83 | 53.11 | 85 | -3,2 | 8,4 | -0,8 | 17,1 | 568 | 51 | 188 |
| BH | 12.32 | 54.12 | 58 | -2,1 | 8,0 | 0,2 | 15,9 | 588 | 48 | 191 |
| GR | 14.73 | 53.32 | 114 | -3,8 | 8,2 | -1,4 | 17,0 | 568 | 57 | 189 |
| KA | 18.14 | 54.24 | 252 | -5,9 | 5,9 | -3,8 | 14,8 | 621 | 107 | 218 |
| KO | 18.43 | 54.25 | 200 | -5,5 | 5,6 | -3,4 | 14,4 | 593 | 99 | 215 |
| NZ | 13.14 | 53.39 | 117 | -2,9 | 7,8 | -0,6 | 16,3 | 580 | 53 | 193 |
| WE | 18.08 | 54.72 | 59 | -4,2 | 7,0 | -2,2 | 15,9 | 623 | 82 | 204 |


Table 2: Initial nutrient concentrations (µg kg DM; mean ± SD) and gravimetric soil moisture at the start of the FTC treatment.

| Site | $NO_3^-$-N | $NH_4^+$-N | $PO_4^{2-}$-P | SM (%) |
|---|---|---|---|---|
| BB | 15.0±0.5 | 14.6±0.9 | 0.09±0.01 | 19,4 |
| BH | 15.0±0.7 | 11.8±1.2 | 0.09±0.02 | 28,1 |
| GR | 9.1±0.3 | 67.5±11.0 | 0.14±0.04 | 27,2 |
| KA | 6.0±1.8 | 55.3±8.2 | 0.60±0.64 | 36,4 |
| KO | 14.8±0.8 | 28.1±2.3 | 0.30±0.28 | 31,6 |
| NZ | 9.7±2.7 | 15.8±1.6 | 0.12±0.02 | 19,6 |
| WE | 1.2±0.3 | 25.5±1.1 | 0.48±0.52 | 36,6 |



## 2.2 *FTC treatment*

The FTC treatment was set up as a fully factorial combination of sample site, FTC magnitude, and FTC frequency in a gradient
design consisting of seven sites along a gradient of winter climate (see above), seven FTC magnitudes (set up equidistantly
between: T = -1.2 and -12°C), and seven FTC-frequencies (f = 1-7). In addition, three control samples without FTC (T = 4.0°C
and f = 0) were analyzed at the end (day 8) of the experiment for each site respectively. In total, this resulted in 364 samples
(7 sites x 7 FTC magnitudes x 7 FTC frequencies + 7 x 3 controls). Gradient experiments with unique (unreplicated) sampling
at each factorial combination have recently been shown to outperform classical, replicated designs in terms of detecting and
characterizing potentially non-linear ecological response surfaces of interacting environmental drivers (Kreyling et al., 2018).
Such designs profit from expanding the range of environmental drivers and are therefore recommended to include extreme and
rather unrealistic values such as the maximum FTC magnitude in our example. Soil temperatures of -12°C rarely occur in
temperate forests. However, they can help elucidating response patterns and might even become possible as future warming
of the Polar Ocean might increase advection of polar air masses, potentially causing unprecedented cold extremes over Europe
(Petoukhov and Semenov, 2010; Yang and Christensen, 2012).

The simulated FTC followed typical FTC for temperate ecosystems with daily cycles between thawed and frozen states. The
FTC treatment was realized for all samples in parallel in programmable climate chambers (Percival LT-36VLX, Percival
Scientific Inc., Perry/Iowa). One FTC lasted 24 h with 2 h at the preset minimum temperature and 12 h at +1°C (sufficient for
thawing but too cold for considerable microbial activity). The rates of temperature change consequently differed between FTC
magnitudes but was < 3 K h$^{-1}$ even for the coldest magnitude. Temperature directly at the soil samples was monitored
throughout the treatment period (7 sensors per FTC magnitude; LogTag TRIX 8, LogTag Recorders Lt, Auckland, New
Zealand) and the realized minimum temperatures per treatment rather than the preset temperature of the climate chambers
were used for further analysis. Directly after the planned FTC frequency was reached for each sample, nutrient extraction and
the subsequent chemical analysis started.

## 2.3 *Nutrient extraction and chemical analysis*

Samples were shaken in 50 ml KCl solution (0.5 M) for 1 h and subsequently filtered through filter paper of 2-3 µm pore size.
Afterwards, the filtrates were stored frozen at -20 °C upon further analysis.

Nitrate was measured after conversion to nitrite at a cadmium reductor column as an azodye (Hansen and Koroleff, 1999).
Samples had to be diluted with ultrapure water (Purelab Flex, Elga) by 50 times. Nitrite was not measured, because its
concentration was expected to be <10 % of nitrate. The nitrate named data are, therefore, the sum of nitrate and nitrite ($NO_x$).
The samples were measured in a segmented flow analyser (FlowSys, Alliance Instruments) equipped with a 5 cm cuvette
(Armstrong et al., 1967). Determination limit for nitrate was 0.32 µmol l$^{-1}$ (4.5 µg nitrate N l$^{-1}$). The combined standard
uncertainty was 4.2 % for samples and the 5 µM standards.





Ammonium was measured as an indophenol blue dye photometrically (Hansen and Koroleff, 1999). Samples had to be diluted
by 50-100 times. The samples were measured in the same segmented flow analyser (Kérouel and Aminot, 1997). Determination
limit for ammonium was 0.43 µmol l$^{-1}$ (6.0 µg ammonium N l$^{-1}$). The combined standard uncertainty was 7.7 % for samples
and the 5 µM standards.

Phosphate concentrations were measured by the molybdenumblue reaction photometrically (Murphy and Riley, 1962). This
was done in the above mentioned autoanlyser (Malcolme-Lawes and Wong, 1990). The determination limit is 0.05 µmol l$^{-1}$
(1.55 µg phosphate-P l$^{-1}$). The determination limit was slightly higher with 0.1 µmol l$^{-1}$. The combined standard uncertainty
was 4.2 % for samples and the 5 µM standards.

Fresh weight of each sample was determined before the start of the FTC treatment. Based on the relation between dry weight
and fresh weight of a further subsample, dry weight of the samples was calculated and nutrient concentrations are reported in
relation to dry weight.

**2.4 *Statistical analyses***

Hierarchical regression analysis was applied to detect and characterize the underlying response patterns in our threefold
interactive gradient experiment according to the recommendations by Kreyling et al. (2018). In short, the hierarchical
regression analysis accepts a more complex model only if it explains the data better than a simpler model, indicated by lower
AICc and, for nested designs, by significant ANOVA comparing the models. Consequently, the final model of a hierarchical
regression analysis contains only those parameters and interactions which help representing the underlying data, i.e. which are
significant for the interpretation of the data.

We first performed linear regression for each single environmental driver (climatic origin expressed as mean minimum air
temperature at the respective sampling site; FTC magnitude expressed as the minimum temperature experienced during the
FTC treatment; and FTC frequency expressed as the number of FTC). Based on the hypothesized non-linear relationship of
nutrient release with these environmental drivers, we then set up different non-linear candidate models for each environmental
driver individually. We chose models known for their ability to describe a wide variety of ecological and biological processes,
i.e. a saturating model (Michaelis-Menten function) and a sigmoidal model (Gompertz function). We used the model
performance indices AICc (Hurvich and Tsai, 1989) to determine the best model. In case of assessing model performance of
linear models or comparing model performance of nested models, we also used ANOVA to test whether the more complex
model explained variation significantly better than the simpler model. We continued by additive combination of the two best
explaining individual models and kept this new model only if it further increased explained variation (lower AICc and
significant model difference in ANOVA). Likewise, we tested whether addition of interactive terms and the third
environmental driver and all other interactive terms between the three drivers to the previous best model further increased
model quality. All steps and all model formulations are documented in Tables 3-5.

All analyses were performed in R 3.4.3 (R Core Team, 2017). Candidate models were fit to the data using 'nlsLM()' of package
'minpack.lm' version 1.2-1. AICc was quantified using 'AICc()' of package 'AICcmodavg' 2.2-1. The overall best model for





each response parameter was visualized using 'scatter3D()' of package 'plot3D' version 1.1.1 and a correlation between measured nutrient release and predicted nutrient release by the model was used to quantify its goodness of fit.

## 3 Results

### 3.1 *Nitrate*

Variation in initial mobile nitrate concentration was large between sample sites (10.1 µg NO3-N per kg dry matter on average ± 5.2 µg NO3-N standard deviation across site averages). Nitrate concentrations at the end of our three-way gradient experiment followed a sigmoid increase towards colder winter minimum temperatures at the sample's origin, which was further modulated by an interaction with FTC magnitude, an interaction between FTC magnitude with FTC frequency, and the three-

way interaction between mean minimum temperature at origin, FTC magnitude, and FTC frequency (Table 3, model 15.). This model achieved a correlation between measured and predicted nitrate concentrations of 0.46. According to this model, highest nitrate concentrations and highest frost sensitivity occurred for the combination of the coldest site, the strongest FTC magnitude, and the highest FTC frequency (Figure 1) with predicted values of up to 16 µg NO3-N per kg dry matter, i.e. a 2.5-fold increase as compared to the initial nitrate concentration before the start of the experiment at this site (site KA, Table 2).

For this combination, also the maximum measured value was found with nitrate concentrations of 37.3 µg NO3-N per kg dry matter. Single, strong FTC (T = - 11 and f = 1), however, also released above average amounts of nitrate for the warmest site. Lowest nitrate concentrations were found for all sites at the mildest FTC magnitude irrespective of FTC frequency. For mild FTC magnitudes, all sites showed below average nitrate concentrations with highest, still below-average, concentrations for the warmest site.

Individually, neither FTC magnitude nor FTC frequency were able to significantly explain nitrate concentrations, more complex saturating or sigmoid models being indistinguishable from the (non-significant) linear model for both parameters (Table 3 models 1.-6.).

Table 3. Results of the hierarchical regression analysis for nitrate concentrations of beech forest soils to changes in FTC

magnitude ($x_1$), FTC frequency ($x_2$), and climatic origin ($x_3$; expressed as mean minimum winter temperature at origin) at the end of the FTC treatments. Tested are linear, saturating (Michaelis Menten function) and sigmoid (Gompertz function) relationships on the single environmental drivers and their interactions. Bold AICc values indicate best model. AICc in italics indicate best single-factor models. $a_1$ to $a_n$ are the fitted parameters of the respective model.

| Model description | model | AICc | Notes |
|---|---|---|---|
| 1.  *Linear, magnitude ($x_1$) only* | $y = a_1 x_1 + a_2$ | *2424* | *Simplest possible start, lm: p = 0.215* |
| 2.  Saturating, magnitude only | $y = \dfrac{a_1 * x_1}{a_2 + x_1}$ | 2800 | Not better than 1. |
| 3.  Sigmoid, magnitude only | $y = a_1 * e^{-a_2 * e^{-a_3 x_1}}$ | 2426 | Not better than 1. |





| | | | |
|---|---|---|---|
| 4. *Linear, frequency ($x_2$) only* | $y = a_1 x_2 + a_2$ | *2425* | *Simplest possible start, lm: p = 0.537* |
| 5. Saturating, frequency only | $y = \dfrac{a_1 * x_2}{a_2 + x_2}$ | 2485 | Not better than 4. |
| 6. Sigmoid, frequency only | $y = a_1 * e^{-a_2 * e^{-a_3 x_2}}$ | 2427 | Not better than 4. |
| 7. Linear, climatic origin ($x_3$) only | $y = a_1 x_3 + a_2$ | 2422 | Simplest possible start, lm: p = 0.066 |
| 8. Saturating, climatic origin only | $y = \dfrac{a_1 * x_3}{a_2 + x_3}$ | 2383 | Better than 7. |
| 9. *Sigmoid, climatic origin only* | $y = a_1 * e^{-a_2 * e^{-a_3 x_3}}$ | *2362* | *Better than 7. and 8., best single factor model* |
| 10. Sigmoid climatic origin and linear magnitude (additive) | $y = a_1 * e^{-a_2 * e^{-a_3 x_3}} + a_4 x_1$ | 2363 | Taking the best model of the best explaining parameter so far (9.) and adding the best model of the second best explaining parameter (1.) |
| 11. Sigmoid climatic origin and its interaction with magnitude | $y = a_1 * e^{-a_2 * e^{-a_3 x_3}} + a_4 x_3 x_1$ | 2354 | Interaction term instead of single factor in 10. new best model |
| 12. Sigmoid climatic origin and its interaction with magnitude and linear frequency | $y = a_1 * e^{-a_2 * e^{-a_3 x_3}} + a_4 x_1 x_3 + a_5 x_2$ | 2354 | Adding best model of third parameter (4.) to best model so far (11.) not better than 11. |
| 13. Sigmoid climatic origin and its two-way interaction with magnitude and frequency, respectively | $y = a_1 * e^{-a_2 * e^{-a_3 x_3}} + a_4 x_1 x_3 + a_5 x_2 x_3$ | 2356 | Adding interaction term climatic origin x frequency to best model so far (11.) ANOVA: not different from 11. with p = 0.671 not better than 11. |
| 14. Sigmoid climatic origin and its two-way interaction with magnitude and two-way interaction magnitude x frequency | $y = a_1 * e^{-a_2 * e^{-a_3 x_3}} + a_4 x_1 x_2 + a_5 x_2 x_3$ | 2352 | Adding two-way interaction magnitude x frequency to best model so far (11.) ANOVA: marginally different from 11. with p = 0.064 new best model |
| **15. Sigmoid climatic origin and its two-way interaction with magnitude and the three-way interaction (climate origin x frequency x magnitude)** | $\boldsymbol{y = a_1 * e^{-a_2 * e^{-a_3 x_3}} + a_5 x_1 x_2 + a_6 x_1 x_3 + a_7 x_2 x_3 + a_8 x_1 x_2 x_3}$ | 2548 | **Adding three-fold interaction term to best model so far (14.) ANOVA: different from 14. with p = 0.007 best model** |
| 16. Linear magnitude and linear frequency without interaction (additive) | $y = a_1 + a_2 x_1 + a_3 x_2$ | 2425 | Checking interaction between magnitude and frequency ANOVA not different from best single factor model (1.): p = 0.309 |
| 17. Linear magnitude and linear frequency with interaction | $y = a_1 + a_2 x_1 + a_3 x_2 + a_4 x_1 x_2$ | 2425 | Checking interaction between magnitude and frequency ANOVA not different from best single factor model (1. p = 0.487) and additive model (16. p = 0.525) |


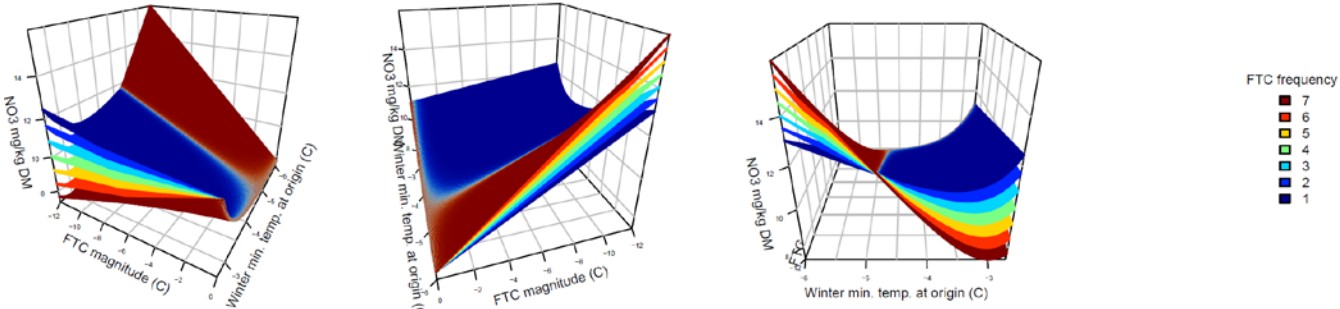

Figure 1: Nitrate concentrations were best explained by the three-fold interactive effects of winter climatic origin (expressed as longterm mean minimum winter temperature at the origin), FTC magnitude (expressed as the minimum temperature

experienced during the FTC manipulation and displayed for freezing temperatures) and FTC frequency during the FTC manipulation. The four dimensional representation is displayed from three different angles (see appendix for an animated version) and is based on the best model fit in the hierarchical regression analysis (model 15. in Table 3 with coefficients $a_1$ = 7.70092; $a_2$ = -22.57795; $a_3$ = 1.52874; $a_4$ = 0.06754; $a_5$ = 0.15402; $a_6$ = 0.03231).

**3.2 *Ammonium***

Variation in initial mobile ammonium concentration was large between sample sites (31.2 µg NH4-N per kg dry matter on average ± 21.7 µg NH4-N standard deviation across site averages). Ammonium concentrations after the FTC treatments followed a sigmoid increase with colder winter minimum temperature at the sample's origin, an additive linear increase with FTC frequency, and were further modulated by an interaction between FTC magnitude with FTC frequency, and the three-

way interaction between mean minimum temperature at origin, FTC magnitude, and FTC frequency (Table 4, model 15.). This model achieved a correlation between measured and predicted ammonium concentrations of 0.61. According to this model, highest ammonium concentrations and highest frost sensitivity occurred for the combination of the coldest site, the strongest FTC magnitude, and the highest FTC frequency (Figure 2) with predicted values of up to 60 µg NH4-N per kg dry matter, i.e. a 10 % increase as compared to the initial ammonium concentration before the start of the experiment at this site (site KA,

Table 2). For this combination, also the maximum measured value was found with ammonium concentrations of 149.7 µg NH4-N per kg dry matter. At this site, FTC frequency had its highest and positively modulating effect while almost no effect of FTC frequency was found for mild FTC magnitude across all origins. Predicted ammonium concentrations and sensitivity to frost decreased rapidly towards the warmer sites with the inflection point of the sigmoid shape at around -3°C for high FTC





magnitudes and -2°C for mild FTC magnitudes. Lowest ammonium concentrations were predicted for the warmest site almost
irrespective of FTC magnitude and FTC frequency.

Individually, FTC frequency, but not FTC magnitude, were able to significantly explain ammonium concentrations, more
complex saturating or sigmoid models being indistinguishable from the linear model for both parameters (Table 4 models 1.-
6.). Their interaction appeared relevant and non-additive (Table 4 models 16. and 17.)

Table 4. Results of the hierarchical regression analysis for ammonium concentrations of beech forest soils to changes in FTC
magnitude ($x_1$), FTC frequency ($x_2$), and climatic origin ($x_3$; expressed as mean minimum winter temperature at origin) at the
end of the FTC treatments. Tested are linear, saturating (Michaelis Menten function) and sigmoid (Gompertz function)
relationships on the single environmental drivers and their interactions. Bold AICc values indicate best model. AICc in italics
indicate best single-factor models. $a_1$ to $a_n$ are the fitted parameters of the respective model.

| Model description | model | AICc | Note |
|---|---|---|---|
| Linear, magnitude (x1) only | $y = a_1 x_1 + a_2$ | 3092 | Simplest possible start, lm: p = 0.182 |
| Saturating, magnitude only | $y = \dfrac{a_1 * x_1}{a_2 + x_1}$ | 3510 | Not better than 1. |
| Sigmoid, magnitude only | $y = a_1 * e^{-a_2 * e^{-a_3 x_1}}$ | 3096 | Not better than 1. |
| Linear, frequency (x2) only | $y = a_1 x_2 + a_2$ | 3088 | Simplest possible start, lm: p < 0.015 |
| Saturating, frequency only | $y = \dfrac{a_1 * x_2}{a_2 + x_2}$ | 3155 | Not better than 3. |
| Sigmoid, frequency only | $y = a_1 * e^{-a_2 * e^{-a_3 x_2}}$ | 3088 | Not better than 3. |
| Linear, climatic origin (x3) only | $y = a_1 x_3 + a_2$ | 2967 | Simplest possible start, lm: p < 0.001 |
| Saturating, climatic origin only | $y = \dfrac{a_1 * x_3}{a_2 + x_3}$ | 2965 | Better than 7. |
| Sigmoid, climatic origin only | $y = a_1 * e^{-a_2 * e^{-a_3 x_3}}$ | 2954 | Better than 8., best single factor model |
| Sigmoid climatic origin and linear frequency (additive) | $y = a_1 * e^{-a_2 * e^{-a_3 x_3}} + a_4 x_2$ | 2946 | Taking the best model of the best explaining parameter so far (9.) and adding the best model of the second best explaining parameter (4.) New best model |
| Sigmoid climatic origin and linear frequency (with interaction term) | $y = a_1 * e^{-a_2 * e^{-a_3 x_3}} + a_4 x_2 + a_5 x_3 x_2$ | 2948 | Adding an interaction term to 10. ANOVA: not different from 10. with p =0.570 |
| Sigmoid climatic origin and linear frequency and linear magnitude | $y = a_1 * e^{-a_2 * e^{-a_3 x_3}} + a_4 x_2 + a_5 x_1$ | 2947 | Adding best model of third parameter (1.) to best model so far (10.) ANOVA: not different form 10. with p = 0.219 |
| Sigmoid climatic origin and linear frequency and interaction climatic origin x magnitude | $y = a_1 * e^{-a_2 * e^{-a_3 x_3}} + a_4 x_2 + a_5 x_1 x_3$ | 2946 | Adding interaction term climatic origin x magnitude to best model so far (10.) ANOVA: not different from 10. with p = 0.219 |




| | | | |
|---|---|---|---|
| Sigmoid climatic origin and linear frequency and interaction frequency x magnitude | $y = a_1 * e^{-a_2*e^{-a_3 x_3}} + a_4 x_2 + a_5 x_1 x_2$ | 2939 | Adding interaction magnitude x frequency to best model so far (10.) ANOVA: different from 10. with p = 0.002 New best model |
| Sigmoid climatic origin and linear frequency and two-way interaction frequency x magnitude and three-way interaction | $\boldsymbol{y = a_1 * e^{-a_2*e^{-a_3 x_3}} + a_4 x_2 + a_5 x_1 x_2 + a_6 x_1 x_2 x_3}$ | 2937 | Adding three-fold interaction term to best model so far (14.) ANOVA: different from 14. with p = 0.025 best model |
| Linear frequency and linear magnitude without interaction (additive) | $y = a_1 + a_2 x_2 + a_3 x_1$ | 3090 | Checking interaction between magnitude and frequency ANOVA different from best single factor model (4.): p = 0.031 |
| Linear frequency and linear magnitude with interaction | $y = a_1 + a_2 x_2 + a_3 x_1 + a_4 x_1 x_2$ | 3083 | Checking interaction between magnitude and frequency ANOVA different from best single factor model (4. p = 0.001) and additive model (16. p = 0.003) |


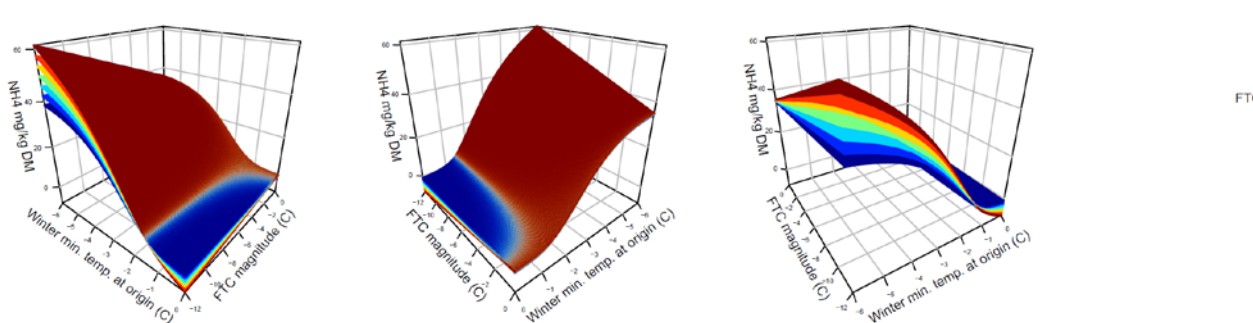

Figure 2: Ammonium concentrations were best explained by the three-fold interactive effects of winter climatic origin (expressed as longterm mean minimum winter temperature at the origin), FTC magnitude (expressed as the minimum temperature experienced during the FTC manipulation and displayed for freezing temperatures) and FTC frequency during the

FTC manipulation. The four dimensional representation is displayed from three different angles (see appendix for an animated version) and is based on the best model fit in the hierarchical regression analysis (model 15. in Table 4 with coefficients $a_1 = 35.77052$; $a_2 = 9.00972$; $a_3 = 0.94421$; $a_4 = 0.06278$; $a_5 = 0.10997$; $a_6 = 0.07065$).





**3.3 *Phosphate***

Variation in initial mobile phosphate concentration was large between sample sites (0.25 µg PO4-P per kg dry matter on average ± 0.21 µg PO4-P standard deviation across site averages). Phosphate concentrations after the FTC treatment followed a sigmoid increase with colder winter minimum temperature at the sample's origin, modulated by an interaction with FTC magnitude, and the three-way interaction between mean minimum temperature at origin, FTC magnitude, and FTC frequency

(Table 5, model 15.). This model achieved a correlation between measured and predicted phosphate concentrations of 0.49. According to this model, highest ammonium concentrations occurred for the combination of the coldest site, the strongest FTC magnitude, and the highest FTC frequency (Figure 3) with predicted values of up to 2.2 µg PO4-P per kg dry matter, i.e. almost a four-fold increase as compared to the initial phosphate concentration before the start of the experiment at this site (site KA, Table 2). The highest measured value for the coldest site was 4.60 µg PO4-P per kg dry matter while the absolute maximum

measured occurred for the strongest FTC magnitude and the highest FTC frequency at site WE (6.70 µg PO4-P per kg dry matter). At the coldest sites, FTC frequency also had its highest and positively modulating effect while almost no effect of FTC frequency was found for mild FTC magnitude across all origins. Predicted phosphate concentrations decreased rapidly towards the warmer sites with the inflection point of the sigmoid shape at around -3°C for high FTC magnitudes and -5°C for mild FTC magnitudes. Lowest phosphate concentrations were predicted for the warmest site with no visible modulation by

FTC magnitude and FTC frequency.

Individually, FTC magnitude, but not FTC frequency, was able to significantly explain phosphate concentrations, more complex saturating or sigmoid models being indistinguishable from the linear model for both parameters (Table 5 models 1.-6.). Their interaction appeared relevant and non-additive (Table 5 models 16. and 17.)

Table 5. Results of the hierarchical regression analysis for phosphate concentrations of beech forest soils to changes in FTC magnitude ($x_1$), FTC frequency ($x_2$), and climatic origin ($x_3$; expressed as mean minimum winter temperature at origin) at the end of the FTC treatments. Tested are linear, saturating (Michaelis Menten function) and sigmoid (Gompertz function) relationships on the single environmental drivers and their interactions. Bold AICc values indicate best model. AICc in italics indicate best single-factor models. $a_1$ to $a_n$ are the fitted parameters of the respective model.

| Model description | model | AICc | Note |
|---|---|---|---|
| 1. *Linear, magnitude ($x_1$) only* | $y = a_1 x_1 + a_2$ | *998* | *Simplest possible start, lm: p < 0.001* |
| 2. Saturating, magnitude only | $y = \dfrac{a_1 * x_1}{a_2 + x_1}$ | 1100 | Not better than 1. |
| 3. Sigmoid, magnitude only | $y = a_1 * e^{-a_2 * e^{-a_3 x_1}}$ | *991* | Best magnitude-only model |
| 4. *Linear, frequency ($x_2$) only* | $y = a_1 x_2 + a_2$ | *1025* | *Simplest possible start, lm: p = 0.369* |
| 5. Saturating, frequency only | $y = \dfrac{a_1 * x_2}{a_2 + x_2}$ | 1028 | Not better than 4. |
| 6. Sigmoid, frequency only | $y = a_1 * e^{-a_2 * e^{-a_3 x_2}}$ | 1028 | Not better than 4. |





| | | | |
|---|---|---|---|
| 7. Linear, climatic origin ($x_3$) only | $y = a_1 x_3 + a_2$ | 986 | Simplest possible start, lm: p < 0.001 |
| 8. Saturating, climatic origin only | $y = \dfrac{a_1 * x_3}{a_2 + x_3}$ | 993 | Not better than 7.. |
| 9. Sigmoid, climatic origin only | $y = a_1 * e^{-a_2 * e^{-a_3 x_3}}$ | 954 | Better than 7., best single factor model |
| 10. Sigmoid climatic origin and sigmoid magnitude (additive) | $y = a_1 * e^{-a_2 * e^{-a_3 x_3}} + a_4$ $* e^{-a_5 * e^{-a_6 x_1}}$ | 955 | Taking the best model of the best explaining parameter so far (9.) and adding the best model of the second best explaining parameter (3.) |
| 11. Sigmoid climatic origin and its interaction with magnitude | $y = a_1 * e^{-a_2 * e^{-a_3 x_3}} + a_4 x_3 x_1$ | 940 | Adding interaction instead of single effect of magnitude to 9. <br> ANOVA: different from 11. with p < 0.001 <br> New best model |
| 12. Sigmoid climatic origin and its interaction with magnitude and linear frequency | $y = a_1 * e^{-a_2 * e^{-a_3 x_3}} + a_4 x_3 x_1$ $+ a_5 x_2$ | 940 | Adding best model of third parameter (4.) to best model so far (11.) <br> ANOVA: not different from 11. with p = 0.128 |
| 13. Sigmoid climatic origin and its two-way interactions with magnitude and with frequency | $y == a_1 * e^{-a_2 * e^{-a_3 x_3}} + a_4 x_3 x_1$ $+ a_5 x_2 x_3$ | 942 | Adding interaction term climatic origin x frequency to best model so far (11.) <br> ANOVA: not different from 11. with p = 0.802 |
| 14. Sigmoid climatic origin and its interaction with magnitude and interaction magnitude x frequency | $y == a_1 * e^{-a_2 * e^{-a_3 x_3}} + a_4 x_3 x_1$ $+ a_5 x_1 x_2$ | 942 | Adding interaction magnitude x frequency to best model so far (11.) <br> ANOVA: not different from 11. with p = 0.701 |
| **15. Sigmoid climatic origin and interaction climate origin x magnitude and threefold interaction** | $\boldsymbol{y == a_1 * e^{-a_2 * e^{-a_3 x_3}}}$ $\boldsymbol{+ a_4 x_3 x_1}$ $\boldsymbol{+ a_5 x_1 x_2 x_3}$ | **937** | **Adding three-fold interaction term to best model so far (11.)** <br> **ANOVA: not different from 11. with p = 0.044** <br> **best model** |
| 16. Sigmoid magnitude and linear frequency without interaction (additive) | $y = a_1 * e^{-a_2 * e^{-a_3 x_1}} + a_4 x_2$ | 992 | Checking interaction between magnitude and frequency <br> ANOVA: not different from best single factor model (3.): p = 0.837 |
| 17. Sigmoid magnitude and linear frequency with interaction | $y = a_1 * e^{-a_2 * e^{-a_3 x_1}} + a_4 x_2$ $+ a_5 x_1 x_2$ | 990 | Checking interaction between magnitude and frequency <br> ANOVA different from additive model (16. p = 0.036) but not different from best single factor model (3.): p = 0.109 |






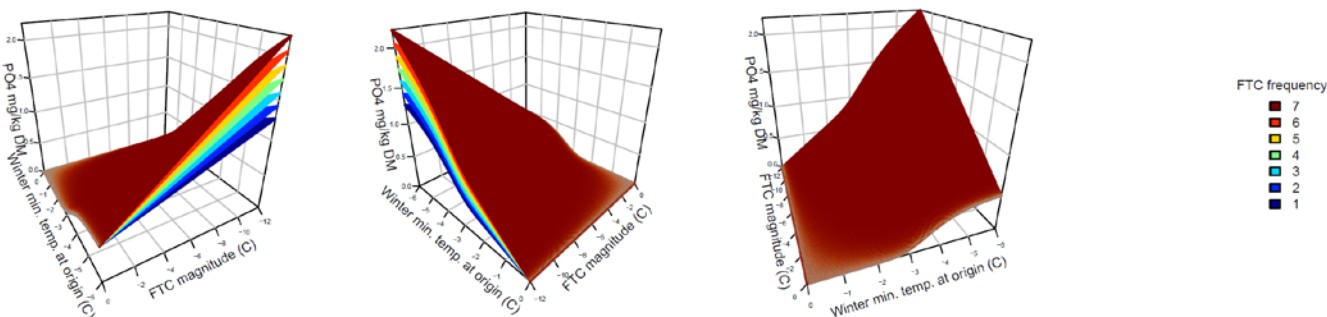

Figure 3: Phosphate concentrations depended on the three-fold interactive effects of winter climatic origin (expressed as longterm mean minimum winter temperature at the origin), FTC magnitude (expressed as the minimum temperature experienced during the FTC manipulation and displayed for freezing temperatures) and FTC frequency during the FTC
manipulation. The four dimensional representation is displayed from three different angles (see appendix for an animated version) and is based on the best model fit in the hierarchical regression analysis (model 15. in Table 5 with coefficients $a_1$ = 0.49455; $a_2$ = 0.01253; $a_3$ = 1.37580; $a_4$ = 0.00890; $a_5$ = 0.00217).

## 4 Discussion

### 4.1 *FTC induce nitrogen release but response patterns are indistinguishable from linear for increased magnitude and*
*increased frequency*

FTC induced short-term nutrient release at high FTC magnitude and frequency in our experiment. Increased nitrate leaching following soil freezing has been explained by decreased root uptake due to lethal or sublethal root damage (Campbell et al., 2014; Matzner and Borken, 2008) and FTC are further reported to increase ammonium production and mineralization rates (Austnes and Vestgarden, 2008; Vestgarden and Austnes, 2009; Shibata et al., 2013; Hosokawa et al., 2017). However, soil
frost commonly reduces nitrification rates and nitrate production (Hosokawa et al., 2017; Hishi et al., 2014; Shibata et al., 2013) as nitrifying bacteria are sensitive to low temperatures (Cookson et al., 2002; Dalias et al., 2002). Based on these aspects, we assume that the processes activating N and P in our experiment are (1) lysis of microbial cells, reported to occur at minimum temperatures of -7°C (Skogland et al., 1988) to -11°C (Soulides and Allison, 1961) or due to osmotic shock upon exposure to melt water (Jefferies et al., 2010), and (2) physical destruction of organic and soil particles (Oztas and Fayetorbay, 2003;
Hobbie and Chapin, 1996) rather than altered mineralization rates as those should be coupled to highest mineral N availability in the unfrozen control. However, FTC increase DON and DOC in temperate deciduous forest soils, quickly leading to enhanced growth of soil microbes and net mineralization, resulting in increased availability of ammonium (Watanabe et al., 2019). Further studies focusing on discrimination between the single processes are clearly needed in light of the strong increases in nitrate (2.5-fold increase) and phosphate (4-fold increase) concentrations over just one week of FTC treatment for
the coldest site and highest FTC magnitudes and frequencies.





Here, we expected to find saturation of nutrient release both with increased FTC magnitude and frequency. However, the observed response patterns of nutrient release along these two drivers were indistinguishable from linear in our experiment. This finding has to be treated with care, though, as both drivers were involved in complex interactions with each other and site of soil origin (see below).

### 335     *4.2 The combination of magnitude and frequency of FTC on nutrient release is not additive*

We assumed FTC magnitude and FTC frequency effects on nutrient release to be additive, but this was not supported by our data. For ammonium, we observed a significant interaction between FTC magnitude and frequency resulting in over-proportionally large release for high magnitude and frequency. However, for all three analyzed nutrients, both these drivers were further involved into significant three-way interactions with site of soil origin and should be interpreted in this sense (see 340   below).

###     *4.3 Soils from colder and snowier forests are more responsive to strong and frequent FTC*

Nutrient release in response to FTC was high for soils from colder and snowier sites. Warmer sites with historically low snow cover showed almost no response to FTC for ammonium and phosphate, while nitrate tended to also be released by strong frost irrespective of FTC frequency in soils from the warmest site. Overall, the strong sigmoidal increase of nutrient 345   concentrations with soils from colder sites was modulated by FTC magnitude and FTC frequency in all studied nutrients. Nitrate concentrations increased with FTC magnitude over the whole range of soil origins peaking for highest frequencies and the coldest sites. The effect of FTC magnitude on ammonium and phosphate concentrations over the climatic gradient was less obvious, but high FTC frequencies mattered only for the coldest sites and high FTC magnitude, then leading to maximum release. All these response shapes show that soils from warmer sites are surprisingly irresponsive to FTC while soils from 350   colder sites are highly sensitive. All studied soils developed under comparable bedrock (sandy Pleistocene deposits) and under the same vegetation types (mono-dominant, mature beech forest with little to no understory). Still, their sensitivity to FTC differed dramatically. Over historic times, the most obvious difference with relevance for FTC sensitivity are winter soil temperature fluctuations, which are small at cold sites characterized by stable, insulating snow cover at the coldest sites and which are large at the warmer sites with their soils over winter being exposed to air temperature fluctuations (Henry, 2008). 355   Higher intensity of FTC changes microbial community composition and functioning, leading to increased tolerance to FTC in temperate forest soils (Urakawa et al., 2014). In light of these results, we suggest that our warmer sites already experienced high winter soil temperature fluctuations with past warming and their microbial community adapted to these conditions, making them comparably irresponsive to our FTC treatments. Contrary, our coldest sites rarely experienced serious FTC in the past, exposing a non-adapted microbial community to FTC stress and leading to high rates of mortality in consequence. 360   These spatial differences in adaptation or legacy of past conditions might also help explaining why microbial responses to mild FTC appear highly divergent with either little to no effects on microbial biomass and nutrient dynamics (Lipson & Monson 1998; Grogan et al. 2004) or temperature fluctuations in FTC down to only -4°C affecting microbial biomass and nutrient

leaching (Larsen et al. 2002; Joseph & Henry 2008). In consequence, largest effects of winter climate change on microbial communities and nutrient dynamics are to be expected for sites where snow cover is currently disappearing (Kreyling, 2020).

The fate of the nutrients released in response to FTC in those regions where snow cover is disappearing is of crucial importance for ecosystem functioning, e.g. tree growth and nitrogen leaching. An increase in available nutrients could increase plant growth. But if the fluctuations in soil temperature lead to lethal or sublethal damage of plant roots (Tierney et al., 2001; Reinmann and Templer, 2018; Kreyling et al., 2012a; Weih and Karlsson, 2002) in parallel to lysis of microbes, the excess nutrients might be leached out of the ecosystem due to reduced root uptake (Matzner and Borken, 2008; Campbell et al., 2014).

The projected increase in winter rain for temperate ecosystems (Stocker, 2014) could then further exacerbate nutrient leaching with the downward flow of the additional water (Bowles et al., 2018).

The applied gradient design analyzed by hierarchical regression analysis (Kreyling et al., 2018) proofed instrumental for the detection and characterization of non-linear response shapes modulated by complex interactions of the environmental drivers. A traditional, replicated design at few treatment levels along the environmental drivers would not have provided these insights

about the complexity of the relationships of the studied drivers.

### 4.4 *Conclusions*

FTC magnitude and, to a lesser extent, also FTC frequency resulted in increased nitrate, ammonium, and phosphate release almost exclusively in soils from cold, snow-rich sites while soils from warmer sites characterized by a history of infrequent snow cover and largely fluctuating soil temperatures were comparably irresponsive to FTC. We propose that currently warmer

forest soils have historically already passed the point of high responsiveness to winter climate change, displaying some form of adaptation either in the soil biotic composition or in labile nutrient sources. This suggests that previously cold sites losing their protective snow cover with climate change are most vulnerable to strong shifts in nitrogen and phosphorus release. In nutrient poor European beech forests of the studied Pleistocene lowlands, nutrients released over winter may get lost when microbes and plant roots are damaged by soil frost and induce reduced plant growth and increased nutrient leaching rates.

## 5. Acknowledgements

We kindly thank the regional forest management (Forst Brandenburg including the Landeskompetenzzentrum Forst Eberswalde, Landesforst Mecklenburg-Vorpommern, National Forest Holding of Poland's State Forests in Szczecin, Gdańsk, and Toruń) for granting access, assistance with site selection and help during the sampling. We are grateful for the help during field sampling, conducting the experiment, and lab analysis by Marcin Klisz, Marc Glaw, Marie E. Meininghaus, and Jonas

Schmeddes. The study was funded by the DFG (German Research Foundation) with grant KR 3309/9-1 and by the DFG research training group RESPONSE (RTG 2010).



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
