# Peer review of "Soils from cold and snowy temperate deciduous forests release more nitrogen and phosphorus after soil freeze-thaw cycles than soils from warmer, snow-poor conditions"

_Biogeosciences, 2020_

## Referee Comment (RC1) · Anonymous Referee #1 · 11 May 2020

General comment

This is a well written manuscript on a timely topic, which studies the effect of freeze thaw cycles on N and P release between snow-poorer warmer and snow-richer colder forest soils. The manuscript is very well and concisely written, clearly structured, and provides clear aims, hypotheses, and approaches. Especially the statistical analysis of the data is very strong, with a transparent use and analysis of the data. The major shortcoming of the study is lack of crucial soil data without whose, the outcome, discussion and conclusion remain speculative.

1. It remains unclear from which soil depth the samples had been taken and what was the criterion of the sampling. Soil temperatures in the mineral soil are known to be well buffered against air temperatures and hence it even remains unclear to which the soils studied have indeed historically experienced FTCs. Although it is described that soil temperatures had been recorded, the data are not shown except minimal winter temperatures.

2. In addition, data about inherent soil properties are lacking. For instance, soil organic matter contents are not given. Soil organic matter is a key soil parameter driving soil microbial communities and the release of nutrients and thus of nutrients released upon lysis of microbial cells. As SOM greatly varies with climate (and which soil depth) it seems likely that it is a key co-variable which could drive the observed responses and as such it should be reported, incorporated into the statistical model and/or discussed.

3. Moreover, the amount of nutrients released should be normed to the amount of nutrients present in the sample. It should be made clear which drivers are/could be responsible for the observed nutrient concentrations: for example inherent nutrient content, C/nutrient ratios that drive the net release of nutrients, or historical FTCs. The concepts and data on other drivers should be considered in the data analysis and/or ruled out in the discussion.

4. Details from the laboratory experiment are lacking or not clear. For example, was the soil moisture kept constant during the incubation, what was the reasoning for the different incubation times or what were the equidistant temperature changes?

Other comments

1. Provide information on effect sizes (increase of nutrient mobilization due to FTC treatments as compared to control). So far, the effect size is only shortly discussed for the most extreme site with the most extreme treatments. And, as far as I understood, for the models the absolute concentrations are used which, however, might be biased by inherent differences. What is the rationale behind the use of absolute concentrations?

2. The reading would be facilitated to have the graphs (Fig. 1-3) at the same spot. This would allow to compare the patterns for the three different nutrients. And in general, use the same perspectives for the graphs and in the same order for all three figures.

3. Visualization of measured analytical data is missing. The graphs from the models are great, but showing the measured data would helpful additional information for many scientists working in this field (or at least provide the information in the supplementary material).

Details/Specific comments:

1. Line 43: "more than"?

2. Line 45: Nutrient limited: Could you be a bit more precise: Which nutrients, is there co-limitation and is it true for all cold-temperate deciduous forests?

3. Line 60: "colder temperatures" than?

4. Line 61: What about cell lysis because of drying-rewetting cycles?

5. Line 62: freeze thaw cycle: abbreviation already introduced

6. Line 65: physiological re-adaptation to thawing conditions may lead to microbial carbon and nutrient release – not sure what this means

7. Line 89: would be interesting to directly give the amount of N deposition in Central Europe as comparison

8. Line 94: P nutrition is recently decreasing: Recently it was researched, but I think the problem is not recent. It is more likely the recent change in C:N:P stoichiometry that can push P to be a limiting factor

9. There are also attempts to analyse biome patterns of FTC effects (e.g. meta-analysis document higher susceptibility of temperate ecosystem than arctic and high latitude; Gao et al., 2017 Global Change Biology)
10. Line 98: FTC has been introduced before

11. Provide information on average snow heights or some measure of FTC frequency available. Mean winter temperature are not necessarily a good indicator for FTC.

12. Line 127: soil sub-samples? Could you be more precise? From which horizon(s) where the soils taken? What was the criterion for soil sampling? This is a very crucial information for nutrient dynamics!

13. Line 135: Information especially about organic matter content but as well of further soil parameters like microbial biomass would be crucial to draw conclusions about the impact of FTC cycles on nutrient mobilization!

14. Line 140: Table 2: Phosphate-ion is three times negatively charged, Column soil moisture (SM): change to English punctuation

15. Line 146: to clarify: equidistantly between -1.2 and -12°C : deltaT= 10.8°C / 7 = 1.542°C T intervals?

16. Line 160: 'Temperature directly at the soil samples'. Please provide details. Same depth, same location? How many FTCs have occurred? Why are the logged data not shown (at least in supplemental information)?

17. Line 163: Just to clarify: samples with 1 FTC were extracted after one day, samples with 5 FTC after 5 days, controls after 7 days? What is the rational behind the immediate sample extraction after the treatment has finished in comparison to the extraction for all samples after 8 days – at the end of all treatments and with the same incubation time?

18. Line 178: molybdenum blue

19. Line 180: Determination limit: Do you mean detection limit?

20. Line 179-180: The determination limit is 0.05 umol L-1 vs. The determination limit was slightly higher with 0.1 umol L-1? Sentences are unclear

21. Line 235: Quality of graphs – resolution, axis labels overlap, axis numbers difficult to read

22. Line 235: Graphs: NO3 data: is this the additional release of NO3 compared to control or just the total NO3 release? Please clarify – and I would suggest to use the numbers normed to the control data

23. Line 270: Table 4: Would be nice to have the models numbered as done in table 3

24. Line 286: copy-paste error? Should probably be phosphate instead of ammonium

25. Line 291: coldest site or sites?

26. Line 316: How do we know its short-term? Like a flush? If there was only one measurement after the treatment?

27. Line 322: activating N and P? wording

28. Line 322: (1) minimum temperatures of -7 to -11°C were only reached for half of the treatments, but increase of nutrient concentration seems to increase linearly with increase FTC magnitude. . . which is also shown with the models with only magnitude as single factor where the linear model was the best. Would it not be expected to see a stronger increase of nutrient release when reaching the -7°C if this explanation (1) is right?

29. Line 358: 'Contrary, our coldest sites rarely experienced serious FTC in the past'. This seems likely but are there any data supporting this statement? As soil had been sampled in well-buffered subsoils, the FTC frequency and magnitude is open

---

## Referee Comment (RC2) · Anonymous Referee #2 · 22 May 2020

The manuscript addresses the effects of FTC magnitude and frequency on the short-term release of nutrients by conducting a three-factorial gradient experiment. Although the experimental design is simple, the hierarchical regression analysis was applied to detect the underlying response patterns in the threefold interactive gradient experiment. Therefore, the manuscript is more innovative from the perspective of analytical methods. I think that the manuscript is particularly well-written. The figures are excellent and do a great job of summarizing your results. Here are some minor suggestions.

Abstract Line 14: Generally speaking, we use "intensity" instead of "magnitude"

[Figure]

Line 20: change "higher frost" to "higher FTC"

Line 29-30: The unit representation is incorrect, there is no subscript and superscript. Please check the full text.

1. Introduction Line 55: delete "(FTC)"

Line 94-96: Compared to nitrogen, there is less description of phosphorus. Could you add more descriptions about phosphorus.

2. Materials & methods Line 104: Can you clarify what you mean by "FTC magnitude"; delete the second "FTC";

Line 128: The collection date and depth of soil samples are not clarified. Soils sampled in different seasons have different properties, such as soil water content, soil microbial biomass, soil nutrients, and so on. Soil microorganisms also show different tolerance to changing temperature in different seasons. So, the unrealistic time of soil collection will affect the experimental results. In addition, why should the soil be stored for 16 weeks before starting the experiment? This may change the original physical and chemical properties of the soil.

Line 130: 10 grams of soil seems to be a bit less, which leads to greater intensity and rate of freeze-thaw cycle than under field conditions.

Table 2: change "$PO_4^{2-}$–P" to "$PO_4^{3-}$–P"; The value of soil moisture is a dot instead of a comma Line 145-146, 158: Could you show the pattern of freeze-thaw cycle with a figure? (Wang, et al., 2015. Effects of freeze-thaw cycles on the soil nutrient balances, infiltration, and stability of cyanobacterial soil crusts in northern China. Plant and Soil (Figure 2))

Line 197-198: Please explain in detail how to use the AICc to determine the best model

Line 232: Please explain the abbreviation of AICc

3. Results Line 239: Explain abbreviations in the legend ("FTC")

Line 261: Change "were" to "was"

4. Discussion Line 315-330: In the paragraph about FTC effects, you discuss potential mechanisms leading to increases in inorganic N and P. The whole discussion did not involve the discussion about phosphorus. Could you add some discussion about phosphorus.

Line 336: delete the second "FTC"

---

## Author Comment (AC1) · 9 Jun 2020

General comment This is a well written manuscript on a timely topic, which studies the effect of freeze thaw cycles on N and P release between snow-poorer warmer and snow-richer colder forest soils. The manuscript is very well and concisely written, clearly structured, and provides clear aims, hypotheses, and approaches. Especially the statistical analysis of the data is very strong, with a transparent use and analysis of

the data. The major shortcoming of the study is lack of crucial soil data without whose, the outcome, discussion and conclusion remain speculative.

Reply: Thank you very much for this positive feedback and the thoughtful and constructive critique! Upon reading your detailed comments, we agree that additional information on soil parameters improves the interpretation of the presented results. Please compare to the replies to your specific points below for details what we have now added to the manuscript.

1. It remains unclear from which soil depth the samples had been taken and what was the criterion of the sampling. Soil temperatures in the mineral soil are known to be well buffered against air temperatures and hence it even remains unclear to which the soils studied have indeed historically experienced FTCs. Although it is described that soil temperatures had been recorded, the data are not shown except minimal winter temperatures.

Reply: To our own surprise, we indeed missed to report the sampling depth and have added this now (line 139: 0-10 cm soil depth) – thanks for spotting this! We have furthermore clarified the soil sampling design there. We further agree that the history of FTC per site is clearly relevant for the interpretation of the site effects. Our statement of recorded soil temperatures was related to the climate chamber trials while all data provided in Table 1 of the initial submission was based on gridded climate data, which generally does not include reliable soil temperature data. However, we have now added on-site soil temperature measurements for four consecutive years across three winters (2016-2019). We acknowledge that this is a short period of observations which does not shed light on historic patterns and changes, but we assume that this recent snapshot has additional value for the interpretation of our results. As already assumed in the first submission, this additional data now suggests that FTC at the studied soil depth are rare at the western end of the gradient with the warmest winter air temperatures. More importantly, FTC are also rare at the coldest sites, very probably due to the high probability of a continuous snow cover insulating the soil against air temperature

fluctuations at these sites (compare to column 'Precipitation as snow' in Table 1). The relevant on-site FTC records have been added to the reworked Table 1 and are picked up again in the discussion (lines 377ff.).

2. In addition, data about inherent soil properties are lacking. For instance, soil organic matter contents are not given. Soil organic matter is a key soil parameter driving soil microbial communities and the release of nutrients and thus of nutrients released upon lysis of microbial cells. As SOM greatly varies with climate (and which soil depth) it seems likely that it is a key co-variable which could drive the observed responses and as such it should be reported, incorporated into the statistical model and/or discussed.

Reply: SOM, pH, and C/N have been added to Table 1 and are now discussed in light of the results (lines. 380ff): "While the soil C/N-ratio appeared irresponsive to the climatic gradient in our study, soil organic matter content increased towards the coldest sites (Table 1). High organic matter content generally increases the susceptibility of soils for nutrient loss with climate change (Liu et al., 2017). Here, we cannot answer how strongly this pattern in organic matter is driven by historic winter soil temperature and occurrence of FTC, but the expectation of increased mineralization with winter soil warming (Gao et al., 2018) would fit to the observed decrease of soil organic matter content with warmer winter climate (Liu et al., 2017). Moreover, the larger pool of organically bound nutrients at the coldest sites may contribute to their observed responsiveness to FTC warming (Gao et al., 2018)."

3. Moreover, the amount of nutrients released should be normed to the amount of nutrients present in the sample. It should be made clear which drivers are/could be responsible for the observed nutrient concentrations: for example, inherent nutrient content, C/nutrient ratios that drive the net release of nutrients, or historical FTCs. The concepts and data on other drivers should be considered in the data analysis and/or ruled out in the discussion.

Reply: This is an important point and we would agree with norming to the amount of

nutrients present if we would only address the impact of the experimental soil temperature manipulation on soil nutrient release rates. However, in our study and in our analysis, we aimed at analyzing the impact of experimental soil temperature manipulation scenarios under explicit consideration of the soil origin. Thus, the soil origin is included in our 3-factorial modelling approach by way of including long-term winter climate variables of the soil origin. Consequently, this accounts for the different initial nutrient contents, as they indeed are characteristic for each soil origin. Standardizing to pure nutrient release rates would, therefore, be contra-productive for our analysis. Still, we agree that this aspect should be picked up in the discussion and have added there (lines 380ff, also cited in the reply above).

4. Details from the laboratory experiment are lacking or not clear. For example, was the soil moisture kept constant during the incubation, what was the reasoning for the different incubation times or what were the equidistant temperature changes?

Reply: Soil moisture was measured directly before the experiment and the samples were kept sealed during the experiment. Based on this, we see no reason to expect a change in soil moisture during the experiment. We have added this information at line 146f. See replies to the detailed comments below for all other aspects mentioned here.

Other comments 5. Provide information on effect sizes (increase of nutrient mobilization due to FTC treatments as compared to control). So far, the effect size is only shortly discussed for the most extreme site with the most extreme treatments. And, as far as I understood, for the models the absolute concentrations are used which, however, might be biased by inherent differences. What is the rationale behind the use of absolute concentrations?

Reply: See our reply to point 3 for the rationale of using absolute concentrations. Concerning effect sizes, we indeed report the increase in nutrient concentrations only for the most extreme values covered as the three-factorial interactions displayed in Figures 1-3 imply that between control conditions (no FTC) and the extremes, any effect size

is possible depending on the factorial combinations of the environmental parameters. We see no option to quantify this multitude of possibilities in one or a few numbers but certainly would be interested in any idea how this could be done!

6. The reading would be facilitated to have the graphs (Fig. 1-3) at the same spot. This would allow to compare the patterns for the three different nutrients. And in general, use the same perspectives for the graphs and in the same order for all three figures.

Reply: The displayed angles of the four-dimensional models (3d plus color code) have been optimized in order to ideally show the response surfaces and their patterns. Unfortunately, we did not find a single perspective and angle which would allow for optimal visibility of the surfaces across all three response parameters. Still, we agree that the displayed views should at least be ordered as comparable as possible and have re-ordered the views to achieve this. In addition, please refer to the animated gifs in the Appendix for the same views across all parameters. While this comparison between the response parameters is of interest, we deem it more important to show each results graphic close to the text where it is presented and therefore do not combine Figures 1-3 into one Figure 1 A-C.

7. Visualization of measured analytical data is missing. The graphs from the models are great, but showing the measured data would helpful additional information for many scientists working in this field (or at least provide the information in the supplementary material).

Reply: We are happy to share our full dataset as supplementary to this paper (see Appendix B_rawdata.csv). We further tried several different options but did not find a convincing way how to display the raw data of our four-dimensional datasets. Unfortunately, all displays of scatterplots in those four dimensions are hardly digestible. Displaying the data in less dimensions, however, is hardly meaningful because of the complex three-factorial interactions (as identified in the hierarchical regression analysis).

Details/Specific comments: 8. Line 43: "more than"?

Reply: compared to their own past. We have added "with climate change" in order to specify this comparison.

9. Line 45: Nutrient limited: Could you be a bit more precise: Which nutrients, is there co-limitation and is it true for all cold-temperate deciduous forests?

Reply: We specifically refer to these points in subchapter 1.3 but have specified the limitation to N and P now already in this very first paragraph (lines 45ff).

10. Line 60: "colder temperatures" than?

Reply: We have added "than $0°C$" for clarity.

11. Line 61: What about cell lysis because of drying-rewetting cycles?

Reply: We acknowledge that drying-rewetting cycles can also cause microbial lysis. In fact, it appears hardly possible to distinguish between drying-rewetting and freezing-thawing as they cause similar effects at the cellular level (with freezing-thawing causing the disappearance of liquid water as stated in the text). Still, we prefer to avoid going into details if the underlying process is driven by 'true' temperature effects such as physical expansion of the freezing water or by the absence of liquid water here. We rather would like to focus on the ecological consequences of altered soil temperatures here in the introduction.

12. Line 62: freeze thaw cycle: abbreviation already introduced

Reply: Thanks for spotting – we are now using the abbreviation here.

13. Line 65: physiological re-adaptation to thawing conditions may lead to microbial carbon and nutrient release – not sure what this means

Reply: Reformulated to "and the physiological re-activation of microbes when soils are thawing can lead to carbon and nutrient release"

14. Line 89: would be interesting to directly give the amount of N deposition in Central Europe as comparison

Reply: added as suggested; 6 to 45 kg N ha−1 year−1 for European beech forests (Rennenberg and Dannenmann, 2015).

15. Line 94: P nutrition is recently decreasing: Recently it was researched, but I think the problem is not recent. It is more likely the recent change in C:N:P stoichiometry that can push P to be a limiting factor

Reply: We agree and have reformulated this statement accordingly: "Linked to the increased growth of forest trees with N deposition, phosphorus (P) nutrition of beech appears to become another limiting factor for beech growth on nutrient poor soils." (L 103ff)

16. There are also attempts to analyses biome patterns of FTC effects (e.g. meta-analysis document higher susceptibility of temperate ecosystem than arctic and high latitude; Gao et al., 2017 Global Change Biology)

Reply: Thanks a lot for pointing out this interesting reference which we now have cited several times in the introduction and discussion.

17. Line 98: FTC has been introduced before

Reply: Thanks for spotting – we are now using the abbreviation here.

18. Provide information on average snow heights or some measure of FTC frequency available. Mean winter temperature are not necessarily a good indicator for FTC.

Reply: We now present soil FTC data measured on site in Table 1. Together with the winter air temperatures and the Precipitation as snow (also Table 1), this data backs up our claim of few FTC at both ends of the climatic gradient due to warm air temperature in the west and due to consistent snow cover in the east.

19. Line 127: soil sub-samples? Could you be more precise? From which horizon(s)

where the soils taken? What was the criterion for soil sampling? This is a very crucial information for nutrient dynamics!

Reply: We agree and are surprised ourselves that this info was not given in the first submission: Per subsample, one soil core from 0–10 cm into the A horizon was taken. The whole soil sampling is now described in more detail in the methods section (lines 132ff).

20. Line 135: Information especially about organic matter content but as well of further soil parameters like microbial biomass would be crucial to draw conclusions about the impact of FTC cycles on nutrient mobilization!

Reply: The requested information has been added to Table 1. In short, soil substrate was silty sand in all cases with pHCaCl2 ranging 3.1–3.5 and organic matter content ranging 3.7–8.5% in the A horizon.

21. Line 140: Table 2: Phosphate-ion is three times negatively charged, Column soil moisture (SM): change to English punctuation

Reply: We are sorry for these inconsistencies and have corrected them here and checked the whole paper for further incidents.

22. Line 146: to clarify: equidistantly between -1.2 and -12_C : deltaT= 10.8_C / 7 = 1.542_C T intervals?

Reply: Indeed, this was the plan. However, we have now added the characterization of the single levels based on the temperatures measured during the experimental treatments in the climate chambers directly at the samples, as they were less equidistant than expected. Please note that all analyses were run with the directly measured values as stated further down in the methods section (Lines 175ff).

23. Line 160: 'Temperature directly at the soil samples'. Please provide details. Same depth, same location? How many FTCs have occurred? Why are the logged data not shown (at least in supplemental information)?

Reply: We believe that this was a misunderstanding: Here we reported the temperature measurements during the treatments in the climate chambers, not measurements at the sites of origin. The climate chamber measurements were used for all analyses. We have now added more details about the temperature measurements in the climate chamber ("Temperature was monitored for each of the 49 frequency x magnitude treatment combinations ... with 7 sensors per FTC magnitude, directly at the incubated soil samples" Lines 175ff) and furthermore we provided a figure based on this data as Appendix A in the supplementary in order to unequivocally display the FTC treatments. We agree that soil temperature from the sites of origin would also have great value for the interpretation of the data and have now added FTC measurements over four consecutive years at the sites of origin to Table 1 (compare to point 1).

24. Line 163: Just to clarify: samples with 1 FTC were extracted after one day, samples with 5 FTC after 5 days, controls after 7 days? What is the rational behind the immediate sample extraction after the treatment has finished in comparison to the extraction for all samples after 8 days – at the end of all treatments and with the same incubation time?

Reply: Yes. We decided for this standardized sampling in terms of time after the final FTC for each treatment as otherwise the period between the last FTC and the analysis could interfere with the treatment effects due to e.g. recovery or lagged responses in microbial activity and/or community composition after frost exposition.

25. Line 178: molybdenum blue

Reply: Space added, thanks for spotting this!

26. Line 180: Determination limit: Do you mean detection limit?

Reply: We specifically quantify the determination limit here, which is the same as the 'limit of quantification' and provides the value from which the concentration was determined with sufficient precision. The detection limit or 'limit of detection' would describe

the value above which the analyte is considered to be detected, i.e. significantly higher than a blank value and is calculated from the standard deviation of blank values with a safety factor (= 3 for 10 blank values). In our case, the detection limit = the determination limit / 3.

27. Line 179-180: The determination limit is 0.05 umol L-1 vs. The determination limit was slightly higher with 0.1 umol L-1? Sentences are unclear

Reply: Indeed, something went wrong here in the initial submission. We have now corrected this, it now reads: "The determination limit was 0.1 $\mu$mol l-1. The combined standard uncertainty was 4.2 % for samples and the 5 $\mu$M standards."

28. Line 235: Quality of graphs – resolution, axis labels overlap, axis numbers difficult to read

Reply: We have now increased the resolution and tried to avoid overlaps between labels and axes. However, the latter is not always possible as we rather optimized the figures to display the response surfaces and in some incidents this leads to axes labels hidden partly behind the graphs. Please also refer to the animated graphics in the appendix.

29. Line 235: Graphs: NO3 data: is this the additional release of NO3 compared to control or just the total NO3 release? Please clarify – and I would suggest to use the numbers normed to the control data

Reply: Please compare to point 3 for our rationale to rely on the absolute concentrations measured rather than standardized data.

30. Line 270: Table 4: Would be nice to have the models numbered as done in table 3

Reply: Indeed, that was the plan – added now.

31. Line 286: copy-paste error? Should probably be phosphate instead of ammonium

Reply: True and corrected, thanks a lot for spotting it!

32. Line 291: coldest site or sites?

Reply: The plural was used intentionally here, but we have now reformulated the sentence in order to improve clarity.

33. Line 316: How do we know its short-term? Like a flush? If there was only one measurement after the treatment?

Reply: True. What was meant was that we measured it directly after the FTC but the formulation was ambiguous and we have now deleted "short-term".

34. Line 322: activating N and P? wording

Reply: Reformulated to "...processes driving the increase in N and P concentrations ...".

35. Line 322: (1) minimum temperatures of -7 to -11_C were only reached for half of the treatments, but increase of nutrient concentration seems to increase linearly with increase FTC magnitude. . . which is also shown with the models with only magnitude as single factor where the linear model was the best. Would it not be expected to see a stronger increase of nutrient release when reaching the -7_C if this explanation (1) is right?

Reply: This is true, thanks for this helpful thought! We have now added this aspect here (now at lines 343ff).

36. Line 358: 'Contrary, our coldest sites rarely experienced serious FTC in the past'. This seems likely but are there any data supporting this statement? As soil had been sampled in well-buffered subsoils, the FTC frequency and magnitude is open

Reply: Please refer to point 1 and the added data in Table 1.

---

## Author Comment (AC2) · 9 Jun 2020

Anonymous Referee #2 The manuscript addresses the effects of FTC magnitude and frequency on the shortterm release of nutrients by conducting a three-factorial gradient experiment. Although the experimental design is simple, the hierarchical regression analysis was applied to detect the underlying response patterns in the threefold interactive gradient experiment. Therefore, the manuscript is more innovative from the perspective of analytical methods. I think that the manuscript is particularly well-written. The figures are excellent and do a great job of summarizing your results.

Reply: Thanks a lot for this positive feedback and your constructive critique below!

1. Here are some minor suggestions. Abstract Line 14: Generally speaking, we use "intensity" instead of "magnitude"

Reply: Both terms have been used in the past and a quick search through the Web of Science does not provide arguments for one or the other in terms of their frequency of occurrence. We have selected the term based on the general framework of disturbance ecology, which characterizes a disturbance by its duration, abruptness, magnitude and frequency (White & Jentsch 2001 Progress in Botany). We now stick to this wording. Not being native English speakers, though, we would be open for good arguments to change the wording.

2. Line 20: change "higher frost" to "higher FTC"

Reply: Changed as suggested.

3. Line 29-30: The unit representation is incorrect, there is no subscript and super-script. Please check the full text.

Reply: Checked and corrected throughout the text.

4. Introduction Line 55: delete "(FTC)"

Reply: As the abbreviation was already introduced in line 43 we now use the abbreviation here and deleted the full term.

5. Line 94-96: Compared to nitrogen, there is less description of phosphorus. Could you add more descriptions about phosphorus.

Reply: True. We have added a few lines on this topic, now at lines 100ff.

6. Materials & methods Line 104: Can you clarify what you mean by "FTC magnitude"; delete the second "FTC";

Reply: We have added a short definition ("the minimum temperature reached during

the freezing phase of an FTC"; line 113).

7. Line 128: The collection date and depth of soil samples are not clarified. Soils sampled in different seasons have different properties, such as soil water content, soil microbial biomass, soil nutrients, and so on. Soil microorganisms also show different tolerance to changing temperature in different seasons. So, the unrealistic time of soil collection will affect the experimental results. In addition, why should the soil be stored for 16 weeks before starting the experiment? This may change the original physical and chemical properties of the soil.

Reply: To our own surprise, we indeed missed to report the sampling depth and have added this now (line 139: 0-10 cm soil depth) – thanks for spotting this! We have also specified the timing of sampling and the experimental treatments there. The timing was based on the rationale that sampling should occur before natural frost events might happen while the treatments were timed for February when typically the most intensive FTC take place in our study area. We have clarified this rationale at lines 139ff.

8. Line 130: 10 grams of soil seems to be a bit less, which leads to greater intensity and rate of freeze-thaw cycle than under field conditions.

Reply: We specifically went for this relatively small amount in order to ensure homogeneous temperature dynamics throughout the samples. Otherwise the buffering effect that you correctly describe would interfere with our FTC treatments in a way that the exact minimum temperature per sample could not be determined, as it would differ within the sample. Such 'controlledness' is the basis of laboratory experiments and with our sample size we can guarantee that the sample conditions (the temperature of the sample) actually reflected the scenarios that were simulated by the climate chambers, because the samples responded almost immediately to the climate chamber temperature. As stated in the text, we aimed at exploring the discrete and causal relationship between soil temperature and nutrient release which can best be investigated with small soil samples in an ex situ approach. We have now added a short

rationale for the relative small sample amounts (line 144).

9. Table 2: change "PO42–P" to "PO43–P"; The value of soil moisture is a dot instead of a comma Line 145-146, 158: Could you show the pattern of freeze-thaw cycle with a figure? (Wang, et al., 2015. Effects of freeze-thaw cycles on the soil nutrient balances, infiltration, and stability of cyanobacterial soil crusts in northern China. Plant and Soil (Figure 2))

Reply: We have corrected the inconsistencies – thanks for spotting! We have furthermore added a graphical representation of the FTC treatments to the Appendix A of the paper for clarification.

10. Line 197-198: Please explain in detail how to use the AICc to determine the best model

Reply: We have added a short explanation at lines 212ff.

11. Line 232: Please explain the abbreviation of AICc

Reply: We have now introduced this abbreviation already at line 212, but have also added it to the table captions.

12. Results Line 239: Explain abbreviations in the legend ("FTC")

Reply: Done.

13. Line 261: Change "were" to "was"

Reply: Done.

14. Discussion Line 315-330: In the paragraph about FTC effects, you discuss potential mechanisms leading to increases in inorganic N and P. The whole discussion did not involve the discussion about phosphorus. Could you add some discussion about phosphorus.

Reply: We have added a short paragraph on implications for N:P imbalance at lines

406ff.

15. Line 336: delete the second "FTC"

Reply: Done.

───────────────────────────────

---

## Author Response (AR2)

Dear Editor,

Thanks a lot for the positive evaluation and smooth handling of this paper! Please find below our replies to the remaining minor issues raised by the reviewer, whom we would like to thank again for constructive remarks and eyes on the details!

Best regards, Juergen Kreyling, Rhena Schumann, Robert Weigel

The authors did an excellent job in incorporating the suggested points and the manuscript has improved a lot and is now considered as acceptable for publication.

Some last picky details:
Line 77: "organic litter" – can there be inorganic litter?

Reply: Valid argument, reformulated to 'organic compounds'.

Line 139: "organic litter layer" – see above

Reply: We have deleted 'organic' here.

Line 150: table caption: "all climatic data is display as" – grammar

Reply: Corrected to ' is displayed'.

Line 150: Unit for organic matter content

Reply: Thanks for spotting this! We have added: '(%)'.

Line 348: "organic and soil particles": strange classification – does this mean that soil particles cannot be organic?

Reply: We have added 'mineral' before 'soil'.

Line 406: "Phosphate is much less mobile in the soil than nitrate and, consequently, leaching of phosphate is not to be expected": -True, phosphate is less mobile than nitrate, but recent studies show, that phosphate can be leached, although in small amounts, but this could have a significant influence on the long-term. This could be one reason for the observed "increasing P limitation with forest age is a global phenomenon" as you write in line 104 - I suggest to attenuate the conclusion in line 410

Reply: We have inserted in line 410: "in absence of phosphorus leaching" in order to attenuate the conclusion.

Line 408: "active N": what is that? Do you mean "reactive"?

Reply: Yes, corrected accordingly.

Still inconsistent introduction and use of abbreviations: N, P … e.g. line 365, line 400

Reply: We have now consistently used 'nitrogen', 'carbon' and 'phosphorus' in the text.